# Appendiceal Endometriosis: A Comprehensive Review of the Literature

**DOI:** 10.3390/diagnostics13111827

**Published:** 2023-05-23

**Authors:** Leila Allahqoli, Afrooz Mazidimoradi, Zohre Momenimovahed, Veronika Günther, Johannes Ackermann, Hamid Salehiniya, Ibrahim Alkatout

**Affiliations:** 1Ministry of Health and Medical Education, Tehran 1467664961, Iran; 2Student Research Committee, Shiraz University of Medical Sciences, Shiraz 7134814336, Iran; 3Department of Midwifery and Reproductive Health, Qom University of Medical Sciences, Qom 3716993456, Iran; 4University Hospitals Schleswig-Holstein, Kiel School of Gynaecological Endoscopy, Campus Kiel, Arnold-Heller-Str. 3, Haus 24, 24105 Kiel, Germany; 5Social Determinants of Health Research Center, Birjand University of Medical Sciences, Birjand 9717853577, Iran

**Keywords:** appendiceal endometriosis, acute appendicitis, endometriosis, appendectomy

## Abstract

**Objective:** the purpose of this review was to evaluate the prevalence of appendiceal endometriosis and the safety of concomitant appendectomy in women with endometriosis or pelvic pain. **Materials and Methods:** We searched the electronic databases Medline (PubMed), Scopus, Embase, and Web of Science (WOS). The search was not subject to any limitation in terms of time or method. The primary research question was: what is the prevalence of appendiceal endometriosis? The secondary research question was: is it safe to perform appendectomy during surgery for endometriosis? Publications that reported data about appendiceal endometriosis or appendectomy in women with endometriosis were reviewed regarding the inclusion criteria. **Results:** We found 1418 records. After review and screening, we included 75 studies published between 1975 and 2021. With regard to the first question of the review, we found 65 eligible studies and divided these into the following two categories: (a) endometriosis of the appendix presenting as acute appendicitis, and (b) endometriosis of the appendix as an incidental finding in gynecological surgery. Forty-four case reports described appendiceal endometriosis in women who were admitted for the treatment of pain in the right-sided lower abdomen. Endometriosis of the appendix was observed in 2.67% (range, 0.36–23%) of women who were admitted due to acute appendicitis. In addition, appendiceal endometriosis was an incidental finding during gynecological surgery in 7.23% of cases (range, 1–44.3%). With regard to the second question of the review, which was the safety of appendectomy in women with endometriosis or pelvic pain, we found 11 eligible studies. Reviewed cases had no significant intraoperative or follow-up complications during the 12 weeks. **Conclusion:** Based on the reviewed studies, coincidental appendectomy appears reasonably safe and was associated with no complications in the cases reviewed for the present report.

## 1. Introduction

Endometriosis is an estrogen-dependent disease defined by the presence of endometrium-like epithelium and/or stroma outside the endometrium and myometrium, usually with an associated inflammatory process [1]. The condition has been reported in 8–10% of adolescents and women of reproductive age [2,3,4]. Common sites of endometriosis are the ovaries, fallopian tubes, pelvic peritoneum, and uterosacral ligaments (also known as the “pelvic” site), whereas the atypical sites of endometriosis include the gastrointestinal tract, urinary tract, soft tissues, and chest (alternatively referred to as the “extra-pelvic” site) [5]. Endometriosis of the appendix has been reported as a rare condition [6], but it may be prevalent depending on the population [7]. In general, the prevalence of appendiceal endometriosis in women with endometriosis and those with deep endometriosis ranges from 0.05% to 1.69% [8,9,10,11] and 2.6% to 13.2%, respectively [12,13]. Endometriosis of the appendix may be asymptomatic or may cause symptoms of acute and chronic appendicitis [14,15]. It is also known to cause cyclic and chronic right lower quadrant pain [7], melena [16], lower intestinal hemorrhage [17,18], cecal intussusception [19,20], and intestinal perforation [20,21].

The preoperative diagnosis of appendiceal endometriosis is challenging. The definitive diagnosis is typically established after a histopathological examination of the appendix [22,23]. The ideal procedure for both, appendectomy and the surgical diagnosis and treatment of endometriosis, is laparoscopic surgery [23,24,25]. According to some authors [12,13], women with endometriosis undergoing surgery should be informed in advance of the potential need for elective appendectomy [26]. Other authors have expressed uncertainty as to whether surgical removal of the appendix is necessary or safe for the treatment of endometriosis [27]. The potential benefits of appendectomy during surgery for endometriosis include the alleviation of chronic pelvic pain, the prevention of future emergent appendectomy, the elimination of appendicitis in patients with endometriosis, and the reduction of costs [28,29]. Furthermore, once the appendix has been removed it becomes easier to distinguish between the exclusion of acute appendicitis (AA) and the exacerbation of endometriosis. However, appendectomy in this setting has been associated with high readmission rates [27]. The present study was performed to clarify existing doubts about the potential benefits or necessity of simultaneous appendectomy during surgery for the gynecological disease. 

## 2. Materials and Methods 

A systematic review was conducted in accordance with the Preferred Reporting Items for Systematic Review and Meta-analyses (PRISMA) guidelines [30]. 

### 2.1. Research Questions

The primary research question was: -What is the prevalence of endometriosis of the appendix in women?

The secondary research question was: -Is concomitant appendectomy safe during surgery for endometriosis and pelvic pain?

### 2.2. Search Strategy and Information Sources

The three databases PubMed/MEDLINE, Scopus, and Web of Science were searched for relevant publications. The following keywords were used for the search in January 2023: “prevalence”, “incidental endometriosis in the appendix”, “appendiceal endometriosis”, “acute appendicitis”, “appendectomy”, “surgical outcomes”, “endometriosis”, and “pelvic pain”. MeSH keywords and Boolean (AND, OR) operators were employed to enhance the selection of entries.

### 2.3. Study Selection

The studies were listed and screened based on the inclusion criteria, using EndNote software (EndNote X9). The selection procedure consisted of three phases: a screening phase, a selection phase, and a data abstraction phase. Three trained authors (LA, AMM, and IA) screened titles and abstracts as part of the review process. Of the titles and abstracts, 403 articles were chosen for a full-text review. Articles that fulfilled the inclusion criteria were included after a checklist-style form had been filled out independently by two writers (LA and IA). Any contradictions in the full-text review were examined by the third author.

### 2.4. Inclusion and Exclusion Criteria

All types of observational studies conducted throughout the world published in the English language and reporting endometriosis of the appendix, confirmed by histological analysis or evaluating the safety of concomitant appendectomy in the surgical management of endometriosis and pelvic pain, were included in the review. No restrictions were imposed in terms of time, study type, or surgery.

We excluded review studies, investigations that reported malignant neoplasia, and also those studies whose findings were not confirmed by histological analysis. The PRISMA flow chart illustrates the process of selection (Figure 1).

### 2.5. Data Extraction and Synthesis

Details of the articles were extracted and reported using a standard form in order to ensure the consistency of this step for all investigations. Data such as the first author, the year of publication, patients’ age, chief complaint, sample size, histology findings, type of surgery, chief indication, outcomes, and design of studies were extracted. Any disagreement was clarified through discussion (with a third external collaborator if necessary). Due to the diverse modes of reporting, we performed a narrative synthesis of the studies.

## 3. Results

### 3.1. Search Results 

A total of 1418 publications, 555 of which were duplicate articles, were found in the various databases. After reviewing the titles and abstracts of the remaining papers, 460 (of 863) were also excluded. Of the remaining articles, 328 were omitted for lack of alignment with the objectives of the study. Finally, the review comprised 75 studies (Figure 1) based on various methodologies: case reports or case series, retrospective and prospective cohort studies, case–control studies, non-randomized interventional studies (quasi-experimental), and cross-sectional surveys. Two studies provided no data about the study design. 

### 3.2. Study Characteristics

Seventy-five studies were deemed eligible for the first question of the review, which was to evaluate the prevalence of endometriosis of the appendix in women. These were divided into the following two categories: (a) endometriosis of the appendix presenting as AA or with symptoms suggestive of chronic or cyclic appendicitis (*n* = 54), or (b) endometriosis of the appendix as an incidental finding in gynecological surgery (*n* = 18). Eleven studies were deemed eligible for the second question of the study, i.e., the safety of appendectomy in the surgical management of endometriosis and pelvic pain (*n* = 11). Eight studies covered the primary as well as secondary questions of the review.

#### 3.2.1. Endometriosis of the Appendix Presenting as AA or with Symptoms Suggestive of Chronic or Cyclic Appendicitis

From 1975 to 2021, 44 case reports described appendiceal endometriosis in women (61 cases) who were admitted for the treatment of pain in the right-sided lower abdomen. These patients were admitted to hospitals with the diagnosis of AA. The average age of the women was 37.9 years and the majority of them had reported at the hospital with characteristic clinical signs and symptoms of appendicitis (generalized or periumbilical abdominal pain that was subsequently identified in the right lower quadrant (RLQ)). All patients underwent surgery for the removal of the appendix. Laparotomy was used in the majority of cases and laparoscopy in the rest. Appendiceal endometriosis was confirmed by histological investigation in all cases. Table 1 provides a summary of published case reports of appendiceal endometriosis.

The prevalence of endometriosis of the appendix presenting as AA was examined in 10 studies (observational case–control (*n* = 1), prospective (*n* = 1), and retrospective (*n* = 8)) between 2010 and 2020. We calculated the overall prevalence of endometriosis in the appendix based on the results of the 10 studies. In all, 5422 women were hospitalized with symptoms of AA, of whom histological tests confirmed appendicitis in 145 (prevalence 2.67%). The prevalence of endometriosis of the appendix in published reports ranges widely from 0.36% to 23%. Details about appendiceal endometriosis in women with the diagnosis of AA are summarized in Table 2.

#### 3.2.2. The Prevalence of Endometriosis of the Appendix as an Incidental Finding in Gynecological Surgery

Several studies have investigated the prevalence of endometriosis of the appendix as an incidental finding in gynecological surgery. In 1983, Nielsen et al. [79] reported that of 21 patients with endometriosis, the appendix was involved in four of these women. According to included studies, the overall prevalence of appendiceal endometriosis was calculated. The prevalence of appendiceal endometriosis was 7.23% in 18 studies comprising 6732 patients who underwent appendectomy after gynecologic surgery for endometriosis or chronic pelvic pain in conjunction with endometriosis (487 of 6732 patients). The prevalence of appendiceal endometriosis in women who had undergone gynecologic surgery ranged widely from 1% [62] to 44.3% [80]. This wide range is justified by various factors. One of the main factors influencing the involvement of the appendix was the severity of endometriosis [13,81,82]. In a retrospective study published in 2021, Ross et al. found that the prevalence of endometriosis of the appendix in women with stage I–II and III–IV endometriosis was 7.0% and 35.2%, respectively [81]. According to a study by Moulder et al. published in 2017, the prevalence of endometriosis of the appendix was 11.6% in women with superficial endometriosis and 39.0% in those with deep endometriosis (DE) [13,80].

The other factors influencing the involvement of the appendix were an abnormal appearance of the appendix and the indication for surgery [81]. When the appearance of the appendix and the indication for surgery were controlled, women with DE had a 5.9-fold greater risk of endometriosis of the appendix compared to women without endometriosis, and a 2.7-fold higher risk of endometriosis of the appendix compared to women with superficial endometriosis [13]. However, the severity of endometriosis was not a risk factor for the involvement of the appendix in Pittaway et al.’s study (1983) [83]. In addition to the above mentioned, the quantity of endometriosis sites is a risk factor for the prevalence of endometriosis of the appendix. In a retrospective analysis published by Ross et al. in 2021, the predicted probability of discovering endometriosis of the appendix rose from 6% in the presence of no positive endometriosis sites to 56% when the patient had four or more sites [81]. The characteristics of studies that assessed the prevalence of endometriosis of the appendix as an incidental finding in gynecological surgery are listed in Table 3.

#### 3.2.3. Safety of Appendectomy in the Surgical Management of Women with Endometriosis and Pelvic Pain

The impact of appendectomy on surgical outcomes in women with endometriosis has been examined in 11 studies comprising 4570 women with endometriosis [76,81,83,84,85,86,93,94,95,96] from 1983 onward. At this time, Pittaway et al. suggested incidental appendectomy at the time of laparoscopy due to the high prevalence of appendiceal endometriosis in women with endometriosis [83]. Cases reviewed in two studies [81,84] had no intraoperative or postoperative complications related to coincidental appendectomy up to 12 weeks postoperatively. Additionally, no statistically significant differences were found between groups in terms of operating time, postoperative changes in hemoglobin levels, duration of hospital stay, postoperative fever or wound infections, blood transfusions, intra-abdominal abscesses, return of bowel activity, complication rates, emergency readmissions, and morbidity [7,76,83,93,95,96]. The studies showed that appendectomy during gynecologic surgery is safe and does not cause any significant associated complications [76,81,83,84,85,86,93,94,95,96]. Table 4 summarizes the findings of studies concerning the impact of appendectomy on surgical outcomes in women with endometriosis. 

## 4. Discussion

It was suggested that, due to the high prevalence of endometriosis of the appendix in women with endometriosis, an incidental appendectomy at the time of surgery may serve as a preventive as well as therapeutic measure [93]. This review was conducted to evaluate: (a) the prevalence of endometriosis of the appendix and (b) the safety of concomitant appendectomy in women with endometriosis or pelvic pain. The results of 75 studies published from 1975 to 2021 were summarized in order to obtain answers to the two questions addressed in the present review. We examined 61 documented cases of appendiceal endometriosis manifested as AA. The chief complaint of the patients was pain in the lower abdomen when menstruating; AA was established upon clinical investigation. 

Endometriosis in the appendix was first reported by von Rokitansky in 1860 [17]. 

Although the theory of retrograde menstruation was proposed as the primary etiologic factor producing endometriosis and affecting the appendix [97], the pathophysiology of endometriosis in the appendix and all extragenital endometriosis is still unknown [98]. Ectopic transplantation via the oviduct has been suggested [98]. Some authors have reported that the ectopic endometrium-like epithelium is generally observed in the muscularis propria or subserosal layer [99], supporting the hypothesis of the pathogenesis of ectopic transplantation [99]. However, some studies revealed that patients with endometriosis in the appendix have no ovarian disease [31,100]. This finding may contradict the hypothesis of ectopic transplantation via the oviduct. Some researchers have suggested other explanations such as epithelial metaplasia or metastatic transplantation [31]. 

Endometriosis in the appendix and periodic menstrual bleeding in the ectopic tissue may trigger AA [46]. Endometriosis of the appendix may be asymptomatic or may cause digestive symptoms (e.g., pain, vomiting, and melena) [98,99]. It is also known to cause cyclic and chronic right lower quadrant pain [7], lower intestinal hemorrhage [17,18], cecal intussusception [19,20], and intestinal perforation [20,21]. It was reported that endometriosis in the appendix may cause digestive symptoms even after menopause [101]. Not only the relationship between digestive symptoms and appendiceal endometriosis remains unclear [31]. Moreover, the association between digestive symptoms caused by ectopic endometriosis in the appendix and periodic menstruation is not known [46,102].

In general, the treatment strategies of endometriosis are determined individually depending on the desire for children [103,104,105], and the stage of disease [104,105]. Endometriosis treatments include supportive care, medication, and surgery [105]. Although AA generally requires prompt surgery [106], which is the gold standard treatment for AA, non-operative management of AA with antibiotics or intravenous fluids and analgesics without antibiotics has been devised as a viable alternative to surgery [107,108]. However, no consensus has been reached on the role of non-operative management in AA [109]. We believe the current review will serve as a timely reminder for gastrointestinal clinicians and general surgeons. 

AA is the most common diagnosis in young women presenting with acute RIF pain at the emergency department [110]. The current review revealed that appendiceal endometriosis occurs in 2.67% of women who undergo appendiceal surgery for RIF pain, and ranges from 0.36% to 23%. This wide range could be the result of the different perceptions of surgeons with respect to the abnormal appendix. In addition, some surgeons may have performed appendectomy only in women with visible endometriosis. This could be attributed to differences in the statistical population, study method, or the results of histological examination. Furthermore, the application of a very detailed histopathology protocol may increase the detection of appendiceal endometriosis from 7.7% to 12.3% [111]. 

A further outcome addressed in the present review was the prevalence of endometriosis of the appendix as an incidental finding in gynecological surgery. In patients who had undergone gynecologic surgery, the prevalence of appendiceal endometriosis was 7.23% (range, 1–44.3%). The involvement of the appendix may contribute to pelvic pain in women with endometriosis. Harris et al. excised appendices of abnormal appearance in women with pelvic endometriosis and chronic pelvic pain. Of 52 excised appendices, 39 (75%) had abnormal histopathology, including appendicitis or periappendicitis, fibrous obliteration, lymphoid hyperplasia, or carcinoid tumor. Endometriosis was confirmed by histopathology in 23% of these appendices [7]. Berker et al. evaluated 231 women who underwent laparoscopic treatment of endometriosis. Appendectomy was performed if the appendix appeared to be abnormal and showed appendiceal adhesions, rigidity, hyperemia, congestion, indurations, or implants of endometriosis (115 women). Histopathologic evidence of appendiceal endometriosis was found in 51 women (44.3%) [80]. In another study, 4 of 5 women with appendices of abnormal appearance had appendiceal endometriosis (80%) [17]. Fayez et al. evaluated the improvement of symptoms after appendectomy in 63 patients with pelvic pain and an appendix of abnormal appearance. Appendiceal endometriosis was found in 6 patients (9.5%). At the 1-year follow-up, 56 patients (89%) reported complete pain relief [112]. In another study, laparoscopic appendectomies for appendices of abnormal appearance in women with endometriosis and chronic pelvic pain were associated with complete pain relief in 60 of 62 patients (97%) [113]. 

The safety of appendectomy in the surgical management of women with endometriosis and pelvic pain was the second question we addressed. We lack any consensus guidelines on opportunistic appendectomy in the management of women with chronic pelvic pain/endometriosis. A few researchers have advocated for elective appendectomies in women with chronic pelvic pain [76,81,83,84,85,86,93,94,95,96]. Agarwala and Liu reported that a routine appendectomy was the only procedure associated with improvement in 91% of women with pelvic pain. These authors included women with appendices of normal appearance. The prevalence of appendiceal endometriosis in this series was 4.4% [91]. As women with endometriosis experience a high risk of re-operation [114], it is the responsibility of gynecological surgeons and researchers to devise appropriate means of minimizing this risk. Although long-term outcomes after appendectomy are lacking [28], the procedure was beneficial in reducing pain in a subset of women with chronic pain in the right lower quadrant [113]. Several decades of gynecologic surgery have shown that appendectomy at the time of primary gynecologic surgery is safe and does not increase the risks of the procedure [81,90,113]. Additionally, several studies have shown that appendectomy does not significantly increase operating times [95,115] given that the substantial body of data confirming the safety and efficacy of opportunistic appendectomy justifies the approach of performing an appendectomy in all women undergoing surgery for endometriosis and pelvic pain. 

## 5. Limitations

The present review consisted of studies in which appendiceal endometriosis was confirmed by histological tests. However, the methods used to confirm the pathology of the appendix were not reported. Since the diagnosis of endometriosis may differ in patients with endometrium-like epithelium and/or stroma, or just one of the two, the reported prevalence of appendiceal endometriosis may have been affected by the histological method used in the studies. 

## 6. Conclusions

Although we lack clear guidelines regarding appendectomy in patients undergoing surgery for endometriosis, it appears that, regardless of the presence or absence of endometriosis, an appendectomy should be taken into consideration in women who have chronic pelvic pain. This is crucial especially when the appendix appears to be abnormal. Prior to surgery, patients should receive appropriate counseling. The surgeon should perform a thorough examination of the appendix during surgery and the operation report must clearly confirm the removal of the appendix.

## Figures and Tables

**Figure 1 diagnostics-13-01827-f001:**
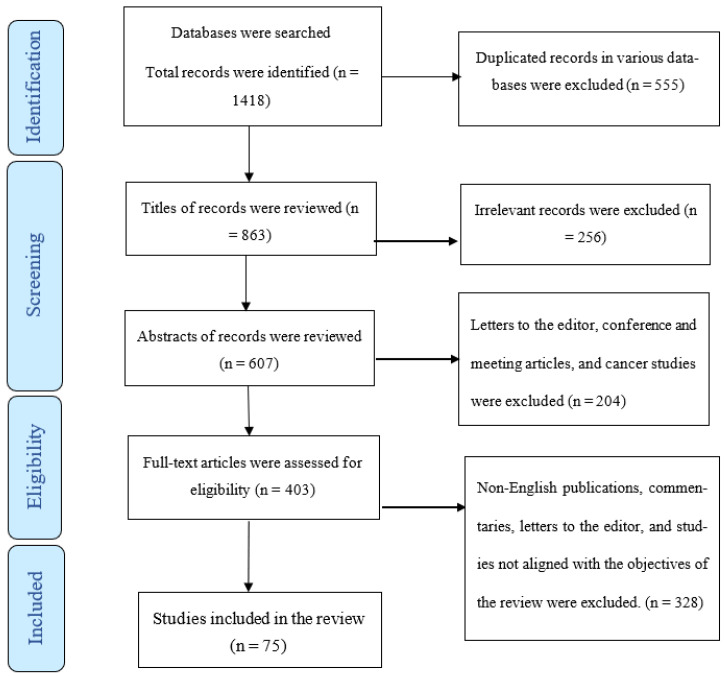
The PRISMA flow chart.

**Table 1 diagnostics-13-01827-t001:** Case reports of endometriosis of the appendix presenting as acute appendicitis or symptoms suggestive of chronic or cyclic appendicitis.

Author/Year	Age (y)	Chief Complaints	Histological Findings
Hori et al., 2021 [31]	38.9	Lower abdominal pain	Appendiceal endometriosis
Reddy et al., 2020 [32]	32	Acute abdominal pain in the RLQ along with nausea and non-bilious vomiting	Endometriosis and intraluminal Enterobius vermicularis
Feldhaus et al., 2020 [33]	45	Right-sided abdominal pain and 10 lb. weight loss over 3 months	Endometriosis of the appendix and cecum
Drumond et al., 2020 [26]	32	Abdominal pain over the last 24 h, which began in the epigastrium and migrated to the right iliac fossa, additional anorexia, nausea, and vomiting	AA caused by appendiceal endometriosis
Perrone et al., 2019 [34]	37	Abdominal pain in the RIF	Appendiceal endometriosis
Jeong et al., 2019 [35]	34	RLQ abdominal pain, nausea, vomiting, headache, constipation, menorrhagia, and dizziness	Appendiceal endometriosis
Gupta et al., 2019 [36]	35	Pain in the RIF and the periumbilical region for the past year	Appendiceal endometriosis
Adeboye et al., 2019 [22]	50	Sharp, non-radiating, intermittent, moderate abdominal pain in the RLQ	Endometriosis involving the appendix without involvement of other pelvic organs
Manoharan et al., 2018 [37]	42	Frequent loose stools over 2 years, elevated calprotectin, and serum C-reactive protein levels	Inverted appendix containing endometriosis
Quirante et al., 2017 [38]	49	Intermittent abdominal pain, initially periumbilical, but migrating to the RLQ	Endometriosis in the colonic wall
Shen et al., 2016 [9]	34	Cyclic RIF pain and an elevated serum C-reactive protein level	Appendiceal endometriosis without macroscopic involvement of other pelvic organs
Mewa Kinoo et al., 2016 [39]	33	Classic clinical signs and symptoms of appendicitis	Normal appendix with endometriosis of the mesoappendix
Wang et al., 2016 [40]	42	Progressive abdominal pain over the RLQ	Endometriosis
Huang et al., 2015 [41]	42	Persistent abdominal pain in the RLQ for 3 days	Appendiceal endometriosis
Dickson-Lowe et al., 2015 [42]	36	Vague periumbilical pain and diarrhea	Appendiceal endometriosis
Ward et al., 2013 [43]	32	LAP and poor appetite which was localized to the RIF	Suggestive of endometriosis
	31	Sudden onset of central abdominal, RIF pain, and vomiting	Appendiceal endometriosis
	45	Sudden onset of central abdominal pain radiating to the right iliac fossa	Foci of endometriosis within its wall
	32	Abdominal pain associated initially with diarrhea and vomiting	Endometriosis
	56	Nausea, diffuse abdominal pain, and loose stools	Appendiceal endometriosis
	41	Chronic dysuria	Appendiceal endometriosis
	22	Headache, nausea, vomiting, diarrhea, abdominal pain localized to the right iliac fossa	Appendiceal endometriosis
Emre et al., 2013 [44]	31	RLQ abdominal pain, which had deteriorated progressively	Endometriosis of the appendix
Ben Maamer et al., 2013 [45]	42	RLQ pain and nausea	Endometriosis of the appendix
Uwaezuoke et al., 2013 [46]	29	Severe RIF pains, tenderness, and rebound tenderness	Appendiceal endometriosis
Malhotra et al., 2012 [47]	37	Pain in the lower abdomen and menorrhagia for the past year	Appendiceal endometriosis
Salati et al., 2011 [48]	29	Appendicitis	Appendiceal endometriosis
Laskou et al., 2011 [49]	25	Abdominal pain in the lower quadrant	Appendiceal endometriosis
Tazaki et al., 2010 [50]	29	RLQ abdominal pain, rebound tenderness localized to McBurney’s point	Appendiceal endometriosis
Sato et al., 2010 [51]	60	LAP with hysterectomy for uterine myoma at age 47	Appendiceal endometriosis
Maghrebi et al., 2010 [52]	19	RLQ pain, nausea, and fever	Appendiceal endometriosis
Astroza et al., 2010 [53]	39	Progressive non-irradiated pain to the RLQ, nauseas, anorexia	Appendiceal endometriosis
Liang et al., 2009 [54]	55	Intermittent right LAP	Appendiceal endometriosis
Azordegan et al., 2009 [55]	37	Right LAP mimicking AA	Ileal endometriosis
Akbulut et al., 2009 [56]	40	High fever, right LAP, and nausea without vomiting	Appendiceal endometriosis
Uncu et al., 2008 [18]	45	Nausea and severe pain spreading throughout the inguinal region	Appendiceal endometriosis
Idetsu et al., 2007 [57]	42	RLQ pain and nausea	Appendiceal endometriosis
Hasegawa et al., 2007 [58]	35	Increasing LAP	Appendiceal endometriosis
Bohlmann et al., 2007 [59]	35	Progressive LAP pain	Appendiceal endometriosis
Tumay et al., 2006 [60]	24	Postprandial pain migrated to the RIF, nausea and vomiting with anorexia	Endometriosis of the appendix vermiformis
Khoo et al., 2006 [15]	40	RIF pain	Appendiceal endometriosis
Shinohara et al., 2001 [61]	50	Occasional abdominal pain and distention	Appendiceal endometriosis
Driman et al., 2000 [62]	34	Cramps in the central abdomen	Proximal appendix showing foci of endometriosis
Ortiz-Hidalgo et al., 1999 [63]	41	Chronic abdominal pain with a tumor in the appendiceal orifice	Endometriosis-like glands or Mullerian duct remnants
Kimura et al., 1999 [64]	41	Abdominal pain and vomiting	Mucosal hyperplasia with mucin-secreting lesions of the appendix
Shome et al., 1995 [65]	33	Painless profuse rectal bleeding	Appendiceal endometriosis
Nopajaroonsri, 1994 [66]	22	Abdominal pain, nausea, and vomiting	Endometriotic obstruction at the distal segment of the appendix
Yelon et al., 1993 [67]	45	Vague and intermittent right-sided abdominal pain	Endometriosis involving fibromuscular tissue and the abdominal wall, the vermiform appendix, and the mesoappendix
Fujisawa et al., 1998 [68]	44	Pain and a mass in the RLQ	Appendiceal endometriosis
Uohara et al., 1975 [69]	28–46 (average: 37)	Generalized abdominal pain	Adenomyosis, endometriosis of the appendix
		None	Endometriosis with bilateral ovarian endometriosis, endometriosis of the appendix and ileum
		LLQ pain during menstruation, occasional constipation	Endometriosis of the left ovary, endometriosis of the appendix
		None	Endometriosis of the appendix
		Dull, constant RLQ pain during menstruation, occasional nausea, vomiting	Leiomyoma of the uterus, endometriosis of the appendix
		Dull, crampy RLQ pain	Leiomyoma of the uterus, endometriosis of the ovaries, endometriosis of the appendix
		Periumbilical pain during menstruation, associated with nausea, vomiting, diaphoresis, diarrhea, flatus	Endometriosis of the right ovary, endometriosis of the serosal surface, endometriosis of the appendix
		Generalized abdominal pain	Adenomyosis, chronic salpingitis, endometriosis of the appendix
		Generalized abdominal cramps, occasional RLQ pain, and diaphoresis	Ovarian endometriosis, bilateral endometriosis of the appendix
		None	Adenomyosis, endometriosis of the ovaries, endometriosis of the appendix
		RLQ pain during menstruation, occasional nausea	Endometriosis of the left ovary, leiomyoma of the uterus, endometriosis of the appendix
		LMQ pain, post-menstrual nausea, vomiting, bloating, gas, constipation during menstruation	Adenomyosis, endometriosis of the left ovary, endometriosis of the appendix

Abbreviations: AA: acute appendicitis; RLQ: right lower quadrant; LLQ: left lower quadrant; RIF: right iliac fossa, LAP: lower abdominal pain, LMQ: left mid quadrant; TAH: total abdominal hysterectomy; SO: salpingo-oophorectomy; BSO: bilateral salpingo-oophorectomy.

**Table 2 diagnostics-13-01827-t002:** Characteristics of studies that reviewed the prevalence of appendiceal endometriosis.

First Author/Year	Sample Size (N)	Chief Indication	Design	Main Results
Coratti et al., 2020 [70]	149	Emergency surgery for RIF pain, undergoing an appendectomy	Observational case–control	Superficial peritoneal endometriosis presented in 23% and 14.7% had endometriosis of the appendix.
Noor et al., 2019 [71]	72	Appendiceal gynecologic proliferations	Retrospective	31.9% endometriosis.
Kinnear et al., 2019 [11]	1214	Emergency appendectomy	Retrospective cohort	11 patients underwent emergency appendectomy and had endometriosis.
Baisakh et al., 2018 [72]	437	Presumptive diagnosis of AA	Prospective	0.25% of cases underwent emergency appendectomy and had endometriosis.
Dincel et al., 2017 [73]	1970	AA	Retrospective	1.7% of patients had endometriosis.
Limaiem et al., 2015 [74]	641	AA	Retrospective	0.2% of cases had rare histopathological findings and underwent emergency appendectomy due to endometriosis.
Emre et al., 2013 [75]	543	AA	Retrospective	Appendiceal endometriosis: 2 cases.
Agrusa et al., 2013 [76]	233	AA	Retrospective	The diagnosis of appendiceal and/or pelvic endometriosis was established in 4.3% of patients.
Shavell et al., 2012 [77]	22	AA	Retrospective case series	4.54% of cases had endometriosis involving the appendiceal wall.
Sieren et al., 2010 [78]	141	AA	Retrospective review	1.41% of patients had endometriosis.

Abbreviations: N: number; RIF: right iliac fossa; AA: acute appendicitis.

**Table 3 diagnostics-13-01827-t003:** Characteristics of studies that reviewed the prevalence of endometriosis of the appendix as an incidental finding in gynecological surgery.

First Author/Year	Sample Size (N)	Chief Indication	Design	The Prevalence of Appendiceal Endometriosis Confirmed by Histological Analysis
Ross et al., 2021 [81]	609	Pelvic pain, stage I–II endometriosis, stage III–IV endometriosis.	Retrospective	Overall, 14.9% (7% and 35.2% in women with stage I–II and stage III–IV endometriosis, respectively).
Nikou et al., 2021 [84]	135	Suspected endometriosis.	Retrospective chart review	18%
Ross et al., 2020 [82]	300	Chronic pelvic pain, stage I–II endometriosis, stage III–IV endometriosis.	Prospective cohort	7.7%
Benoit et al., 2018 [85]	1876	Appendectomies at the time of gynecologic surgery.	Retrospective review	2.1%
Moulder et al., 2017 [13]	395 (151 endometriosis)	Coincidental appendectomy during benign gynecologic surgery.	Retrospective	Overall, 13.2% (11.6% and 39.0% of women with superficial and deep endometriosis, respectively).
Benoit et al., 2017 [86]	741	Pelvic pain and endometriosis.	Retrospective chart review	1.07%
Jocko et al., 2013 [87]	71	Endometriosis, chronic pelvic pain, pelvic mass, others.	Retrospective review	37%
Abrão et al., 2010 [88]	737	Endometriosis confirmed by histology.	Retrospective	1.91%
Song et al., 2009 [89]	772	Infertility, endometriosis, and other diverse pelvic pathologies.	NA	5.2%
Gustofson et al., 2006 [17]	120	RLQ pain and possible endometriosis.	Case series	4.1%
Berker et al., 2005 [80]	231	Chronic pelvic pain in conjunction with endometriosis.	Retrospective chart review	44.3%
Salom et al., 2003 [90]	100 cases 100 controls	Incidental appendectomies at the time of abdominal hysterectomy.	Retrospective case–control	1%
Agarwala et al., 2003 [91]	317	Chronic pelvic pain or endometriosis.	Retrospective review	4.4%
Lyons et al., 2001 [92]	190	Pelvic pain.	Retrospective study	9.47%
Harris et al., 2001 [7]	52	Endometriosis and RLQ pain.	Non-randomized clinical trial	31%
McTavish et al., 1994 [93]	85	Laparoscopy.	Retrospective review	42.4%
Pittaway, 1983 [83]	65 cases, 60 control	Surgery for endometriosis with/without appendectomy.	NA	13%
Nielsen et al., 1983 [79]	21	Endometriosis of the vermiform appendix.	Retrospective	19.04%

Abbreviations: N: number; NA: not available; RLQ: right lower quadrant.

**Table 4 diagnostics-13-01827-t004:** Safety of appendectomy in the surgical management of women with endometriosis and pelvic pain.

First Author/Year	Sample Size (N)	Type of Surgery Laparotomy/Laparoscopy	Design	Outcomes	Main Results
Ross et al., 2021 [81]	609	-	Retrospective	-	No intraoperative or postoperative complications related to coincidental appendectomy were reported up to 12 weeks after surgery.
Nikou et al., 2021 [84]	135	-	Retrospective chart review	-	Appendiceal histopathology was not significantly associated with intra-operative appendiceal characteristics.
Benoit et al., 2018 [85]	1876	Laparoscopy	Retrospective review	Length of stay.	No complications related to the appendectomy procedure itself. The appendectomy cohort had a significantly shorter duration of hospital stay: 1.1 vs. 1.3 days.
Benoit et al., 2017 [86]	920	406 laparoscopic/robot-assisted procedures and 338 exploratory laparotomy procedures.	Retrospective chart review	-	No complications were reported.
Mohling et al., 2016 [94]	79	Laparoscopic surgery	Retrospective analysis	-	No surgical complications related to the appendectomy.
Agrusa et al., 2013 [76]	233	Laparoscopy	Retrospective	Blood loss.	No significant intraoperative blood loss and no blood transfusions. No intraoperative complications, wound complications or intra-abdominal abscesses.
Lee et al., 2011 [95]	356 (appendectomy group, n = 172; no appendectomy group, n = 184).	Laparoscopic surgery	Retrospective	Operating time, return of bowel activity, changes in hemoglobin levels, hospital stay, and any complications.	No statistical differences between groups in regard to operating time, postoperative changes in hemoglobin levels, duration of hospital stay, return of bowel activity, or complication rates.
Leesanghun et al., 2007 [96]	100 (50 cases and 50 controls)	Laparoscopic surgery	Retrospective case-controlled	Duration of the postoperative hospital stay and other postoperative complications.	No significant differences between the two groups with respect to the period of no food ingested orally (1.1 days vs. 1.1 days), the duration of the postoperative hospital stay (5.0 days vs. 4.9 days), and postoperative complications.
Harris et al., 2001 [7]	52		Non-randomized clinical trial	Complication.	No major complications or emergency readmissions after appendectomy.
McTavish et al., 1994 [93]	85	Laparoscopy	Retrospective review	Morbidity.	No immediate or late procedure-related morbidity.
Pittaway, 1983 [83]	125 (65 cases, 60 control	Surgery for endometriosis with/without appendectomy	NA	Days of hospitalization, postoperative fever or wound infection.	No statistically significant difference between the two groups in regard of the days of hospitalization, postoperative fever, or wound infection.

Abbreviations: N: number; NA: not available; RLQ: right lower quadrant.

## Data Availability

The data that support the findings of this study are available from the corresponding author upon reasonable request.

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
