# Peer review of "Appendiceal Endometriosis: A Comprehensive Review of the Literature"

_diagnostics, 2023, doi:10.3390/diagnostics13111827_

Round 1

Reviewer 1 Report

1.      The authors review 61 cases of appendiceal endometriosis in women admitted for right lower abdominal pain.

2.      A study of 61 select cases is inadequate to draw global conclusions. The conclusions need to be specific for the cases studied.

3.      There are no line numbers and text will need to be searched for my comments.

4.      List the inclusion criteria.

5.      List the exclusion criteria.

6.      Was histologic confirmation of both endometriotic glands and stroma required? If only one was required, were stromal and epithelial markers used to aid in diagnosis? If you do not know the answers, list that as a limitation.

7.      How many were based on a surgical diagnosis only?

8.      How many were based on a histologic diagnosis only?

9.      How many were based on a surgical diagnosis with histologic confirmation?

10.  If you do not know the answers to the previous three questions, list that as a limitation.

11.  “the presence of endometrial tissue” has been incorrect since Samson (1921, doi: 10.1001/archsurg.1921.01110080003001) concluded that endometriosis was different from endometrium both "in structure and function." There have been multiple studies that conclude that the source of endometriosis may be retrograde endometrium, Müllerian remnant, metaplastic peritoneum, or disseminated bone marrow stem cells. Unless you have data that shows that all endometriosis is endometrium, please change “endometrial” to "endometrial-like" or "endometriotic." “Endometrioid” is generally reserved for endometriotic cancer. (Clement 2007, doi: 10.1097/PAP.0b013e3180ca7d7b. PMID: 17592255) Giudice (2010) has endometriosis “is characterized by endometrial-like tissue outside the uterus.” Giudice LC. Clinical practice. Endometriosis. N Engl J Med, 2010, 362(25), 2389-2398. doi: 10.1056/NEJMcp1000274. PMID: 20573927. PMCID: PMC3108065.

12.  “Between patients with endometriosis and those with deep endometriosis,…” needs to be “Comparing patients with endometriosis and those with deep endometriosis,…”

13.  “appendiceal endometriosis (AppE)”is not used until the ninth mention and is not used consistently including not in tables 1 and 2, but in Table3. I do not like non-standard abbreviations and would not use “AppE,” but, if you must, then use it consistently.

14.  “There are currently 61 documented cases of appendix endometriosis …” is specific to this study and may not include all cases. Consider “This study examines 61 cases of appendix endometriosis …”

15.  “serve as a timely reminder for general and gastrointestinal.” ends on an adjective. What is the object of the adjective?

16.  I agree with “It's also conceivable that some surgeons restricted appendectomy to patients with endometriosis, which was clearly evident.” In my patients, appendiceal endometriosis was palpable but not seen in 50% of cases. There is a study that shows a decreased surgical documentation of histologically documented endometriosis in patients with coexisting diseases such as myomata, pelvic cancer, and severe adhesions.

17.  “Benefits of appendectomy…” should be “Potential benefits of appendectomy…”

18.  “Result of studies show there are no intraoperative or postoperative complications related to coincidental appendectomy up to 12 weeks postoperatively.” is specific for this study and cannot be generalized to all cases. This is better as “These cases had no intraoperative or postoperative complications related to coincidental appendectomy up to 12 weeks postoperatively.” The corollary is that “based on 61 cases, coincidental appendectomy appears reasonably safe” and “no associated complications” needs to be “no associated complications in 61 cases.”

19.  “Endometriosis of the appendix vermiformis is a rare condition” does not appear reasonable with a 2.7% prevalence and “high occurrence” later in the paper. “Endometriosis of the appendix vermiformis has been reported as a rare condition in some (7), but not all (93), studies.” sounds more reasonable.

1.      Some of my comments may apply to American English and not British English; use them as you see needed. The title and data appeared mismatched until I understood that you were using a convention that is uncommon in America; it may be more common in the UK; that I do not know. Also, this is about cases in populations, not populations. The title is easier for me to read and understand as "Appendiceal endometriosis in cases diagnosed as acute appendicitis: A review study" The implication of that is that eight of the ten uses of “the appendiceal” need “the” removed. It is correctly used in two cases in Table 1. “A population” needs to be changed to “cases” 19 times with the accompanying verbs changed to agree with the plural “cases.”

2.      The data, review, and most conclusions are good. The English is either British or poorly written; this may make it difficult for American English speakers to read. I am an American and do not understand some British conventions. As I do not know if it is good British English, have it read either by a British English expert or rewritten by a professional American English writing service.

3.      There is a mixture of British and English spelling such as the American “gynecologic” and British “haemoglobin.” Use either British or American English consistently.

4.      “who admitted” sounds British. Consider “who were admitted” or “admitted” in three locations.

Author Response

Reviewer#1

1. The authors review 61 cases of appendiceal endometriosis in women admitted for right lower abdominal pain.

2.      A study of 61 select cases is inadequate to draw global conclusions. The conclusions need to be specific for the cases studied.

Thank you for your feedback and comments.

We appreciate your time and valuable opinion.

It was revised.

Page 1, lines 36-38

3.      There are no line numbers and text will need to be searched for my comments.

Sorry for any inconvenience. We added line number.

4.      List the inclusion criteria.

We added inclusion criteria.

Page 3, lines 20-29

5.      List the exclusion criteria.

We added exclusion criteria.

Page 3, lines 20-29

6.      Was histologic confirmation of both endometriotic glands and stroma required? If only one was required, were stromal and epithelial markers used to aid in diagnosis? If you do not know the answers, list that as a limitation.

Thank you, we added this sentence:

“…reporting endometriosis of the appendix confirmed by histological analysis” in conclusion criteria section. And also, due to lack of data on diagnosis of endometriosis based on stromal and epithelial markers, we pointed out this in limitation section of study.

Page 15, limitation section.

7.      How many were based on a surgical diagnosis only?

Endometriosis of the appendix confirmed by histological analysis. Confirmation by histological analysis was inclusion criteria of this study.

8.      How many were based on a histologic diagnosis only?

All included studies were confirmed by histological analysis

9.      How many were based on a surgical diagnosis with histologic confirmation?

All included studies were confirmed by histological analysis

10.  If you do not know the answers to the previous three questions, list that as a limitation.

11.  “the presence of endometrial tissue” has been incorrect since Samson (1921, doi: 10.1001/archsurg.1921.01110080003001) concluded that endometriosis was different from endometrium both "in structure and function." There have been multiple studies that conclude that the source of endometriosis may be retrograde endometrium, Müllerian remnant, metaplastic peritoneum, or disseminated bone marrow stem cells. Unless you have data that shows that all endometriosis is endometrium, please change “endometrial” to "endometrial-like" or "endometriotic." “Endometrioid” is generally reserved for endometriotic cancer. (Clement 2007, doi: 10.1097/PAP.0b013e3180ca7d7b. PMID: 17592255) Giudice (2010) has endometriosis “is characterized by endometrial-like tissue outside the uterus.” Giudice LC. Clinical practice. Endometriosis. N Engl J Med, 2010, 362(25), 2389-2398. doi: 10.1056/NEJMcp1000274. PMID: 20573927. PMCID: PMC3108065.

Thank you for this valuable comment.

We defined the endometriosis based on the An international terminology for endometriosis, 2021.

Page 2, lines 1-4.

12.  “Between patients with endometriosis and those with deep endometriosis,…” needs to be “Comparing patients with endometriosis and those with deep endometriosis,…”

Based on this comment, studies were reviwed again.

We added this sentence.

According to a study by Moulder et al. published in 2017, the prevalence of endometriosis of the appendix was 11.6% in women with superficial endometriosis and 39.0% in those with deep endometriosis (DE) (86).

Page 10, lines 11-13,

13.  “appendiceal endometriosis (AppE)”is not used until the ninth mention and is not used consistently including not in tables 1 and 2, but in Table3. I do not like non-standard abbreviations and would not use “AppE,” but, if you must, then use it consistently.

Thank you for bringing this to our attention

It was revised.

14.  “There are currently 61 documented cases of appendix endometriosis …” is specific to this study and may not include all cases. Consider “This study examines 61 cases of appendix endometriosis …”

Thank you for this comment. The sentence was revised according this comment.

Page 14, lines 10- 11.

15.  “serve as a timely reminder for general and gastrointestinal.” ends on an adjective. What is the object of the adjective?

It was revised.

Page 14, lines 30-31.

16.  I agree with “It's also conceivable that some surgeons restricted appendectomy to patients with endometriosis, which was clearly evident.” In my patients, appendiceal endometriosis was palpable but not seen in 50% of cases. There is a study that shows a decreased surgical documentation of histologically documented endometriosis in patients with coexisting diseases such as myomata, pelvic cancer, and severe adhesions.

Thank you for your feedback and sharing your worthy experience with research team.

17.  “Benefits of appendectomy…” should be “Potential benefits of appendectomy…”

It was revised.

Page 2, lines 27 and 35

18.  “Result of studies show there are no intraoperative or postoperative complications related to coincidental appendectomy up to 12 weeks postoperatively.” is specific for this study and cannot be generalized to all cases. This is better as “These cases had no intraoperative or postoperative complications related to coincidental appendectomy up to 12 weeks postoperatively.” The corollary is that “based on 61 cases, coincidental appendectomy appears reasonably safe” and “no associated complications” needs to be “no associated complications in 61 cases.”

Thank you for bringing it to our attention.

Page 1, lines 36-38

And also,

We revised in in result section:

Cases reviewed in two studies had no intraoperative or postoperative complications related to coincidental appendectomy up to 12 weeks postoperatively.

Page 12, lines 5-6

19.  “Endometriosis of the appendix vermiformis is a rare condition” does not appear reasonable with a 2.7% prevalence and “high occurrence” later in the paper. “Endometriosis of the appendix vermiformis has been reported as a rare condition in some (7), but not all (93), studies.” sounds more reasonable.

It was revised.

Page 2, lines 11-12

Comments on the Quality of English Language

1.      Some of my comments may apply to American English and not British English; use them as you see needed. The title and data appeared mismatched until I understood that you were using a convention that is uncommon in America; it may be more common in the UK; that I do not know. Also, this is about cases in populations, not populations. The title is easier for me to read and understand as "Appendiceal endometriosis in cases diagnosed as acute appendicitis: A review study" The implication of that is that eight of the ten uses of “the appendiceal” need “the” removed. It is correctly used in two cases in Table 1. “A population” needs to be changed to “cases” 19 times with the accompanying verbs changed to agree with the plural “cases.”

Thank you for feedback. Whole manuscript was edited by a native English editor.

2.      The data, review, and most conclusions are good. The English is either British or poorly written; this may make it difficult for American English speakers to read. I am an American and do not understand some British conventions. As I do not know if it is good British English, have it read either by a British English expert or rewritten by a professional American English writing service.

Thank you for feedback. Whole manuscript was edited by a native English editor.

3.      There is a mixture of British and English spelling such as the American “gynecologic” and British “haemoglobin.” Use either British or American English consistently.

Thank you for feedback. Whole manuscript was edited by a native English editor.

4.      “who admitted” sounds British. Consider “who were admitted” or “admitted” in three locations.

Thank you for feedback. Whole manuscript was edited by a native English editor.

Reviewer 2 Report

This article contains a review of several aspects of the co-existence of endometriosis and appendiceal pathology. It also discusses the opportunity to systematically remove the appendix during abdominal surgery.

I believe that, if properly modified, this manuscript will be worth of publication.

However, before publication can be considered, a series of general and specific issues must be addressed.

GENERAL ISSUES

1.       As far I could see, the numbering of pages is missing, and the numbering of lines is only provided for the Discussion. This made the review unduly complicated and quite possibly I may have made errors in numbering lines.

2.       The manuscript is “flooded” with long Tables. I am not sure that this type of details adds to the text, while it alters the layout of the manuscript in several pages, placing the text sideways.

3.       The English text must be improved. I am fully aware of the difficulty encountered by scientists in non-English-speaking countries; nonetheless, the text of an article published in English must be properly written.

To help the Authors, I have made a series of suggestions, but it is the entire manuscript that requires review.

4.       The manuscript mentions surgery for endometriosis, as well as surgery for appendicitis. I recommend that whenever surgery is mentioned, the Authors specify whether they refer to “laparoscopic” or “laparotomic” surgery, or to both.

5.       In the Abstract it is stated that a total of 1418 entries were subjected to “full-text screening”. If this were true, even if an article would take only 10 minutes to read, Authors would have spent more than 235 hours in screening their entries! In addition, under 2.2., Study selection, it is stated that “three trained authors screened titles and abstracts…”. This makes sense and the text of the Abstract should be modified, to coincide with what was done.

6.       In the Introduction (page 2), mention should be made of whether any specific symptoms have been described in women with appendiceal endometriosis.

7.       Authors should be very careful in their description of the various groups to avoid misunderstanding, since the frequency of positive findings depends on how grouping is made. See Item 6 of the Specific Points.

SPECIFIC POINTS

 1.       Abstract: line 1: “This review was conducted to evaluate to presence…”. Lines 15-16: “women who were admitted…”; “The presence of appendiceal endometriosis was observed in 2.67%...”. Line 17: ”women who were admitted to the hospital…”. Line 21: “Conclusion: The presence of appendiceal endometriosis…”

2.       Introduction (page 2): Line three: What exactly do the authors mean when they mention endometriosis of “the genitalia”? Lines 5-6: “can be less frequently…”. Line 11: “must be distinguished…”. Line 18: “the preferred…”. Line 21: “are appendix-related complications…” Line 25: “several researchers…”. Lines 26-27: “disorders affecting women”; this is a vague statement: a brain tumor is a disorder that can affect a woman but does not require appendicectomy. Line 28: Benefits of carrying out an appendicectomy during surgery for…”. Lines 37-38: “in the present review we collected evidence for the topic”.

3.       Materials and methods. Page 2, first paragraph: I believe that authors meant that they have examined the “concomitant presence” of peritoneal and  appendiceal endometriosis. Page 3, line 3: “occurrence” of what? Line 14. It seems to me that what the Authors mean is “ acute appendicitis” (I am unaware of the existence of an “acute endometriosis”). Lines 20-21: “Conducted anywhere in world and published in English. No restrictions were applied to the search.

4.       Results, page 3: Line 9 from the bottom: We initially identified 1418 publications…”. Line 8 from the bottom: What exactly is meant by “duplicate articles”? I presume that they were quoted more than once. Last two lines: were these 2 studies included?

5.       Study characteristics (page 4), line 1: “The present review includes 75 articles…”.

6.       Item 3.2.1., page 6. Line 2: “who were admitted…”. This paragraph states that “appendiceal endometriosis was the most reported histological finding in these women”. There is something missing in this description because the text leads to the conclusion that in most cases of acute appendicitis, this is caused by endometriosis. This cannot be, since authors collected 44 cases over some 45 years, whereas every year there are thousands of cases of acute appendicitis that undergo surgery, as documented in Item 3.2.2. on page 9.

7.       Item 3.2.3., page 10, Line 2: “reported on…”. Line 3: “The incidence of appendiceal endometriosis…”. Line 14: Instead of “while” it should be “On the other hand, the investigation by Pittaway et al concluded that...”. Line 16: I presume that authors mean that “the number of endometriotic foci…”.

8.       Item 3.2.4, page 11, Line 4: “According to these studies…”

9.       Discussion (page 13), first three lines: I am not sure of the meaning of this sentence: what is meant by the words “the high incidence of diseases in women with endometriosis”. Which diseases? Line 4: Obviously, a review is conducted to review a subject! The text needs improvement. Line 19: It is stated that the pathogenesis of appendiceal endometriosis involves retrograde menstruation through the Fallopian tubes. Since the appendix does not communicate with the peritoneal cavity, such a hypothesis needs further explanation. Lines 21-23: the sentence is incomplete. Lines 24.25: this sentence is also incomplete. Lines 28-31: I am not sure I understand why a histological diagnosis would depend on the “perception of the surgeon”. Line 38: The incredible range of frequency mentioned needs an explanation, even tentative.

In my review report I expressely mentioned the need to improve the English text.

Author Response

Reviewer#2

GENERAL ISSUES

1.       As far I could see, the numbering of pages is missing, and the numbering of lines is only provided for the Discussion. This made the review unduly complicated and quite possibly I may have made errors in numbering lines.

Dear reviewer, we appreciate your time and comments.

Sorry for any inconvenience, we made the number lines and page. We are forward to hearing your comments.

2.       The manuscript is “flooded” with long Tables. I am not sure that this type of details adds to the text, while it alters the layout of the manuscript in several pages, placing the text sideways.

Thank you for this valuable comment. Yes, we agree that there are several tables and tables are long. But to answer the questions of this study, we should have comprehensively seen and included the published studies.

We searched to find standard tips on table design, we find the tips below in the literature: 

‘Summary of findings’ tables include the following elements using one of the accepted formats.

1.      A brief description of the population and setting addressed by the available evidence (which may be slightly different to or narrower than those defined by the review question).

2.      A brief description of the comparison addressed in the ‘Summary of findings’ table, including both the experimental and comparison interventions.

3.      A list of the most critical and/or important health outcomes, both desirable and undesirable, limited to seven or fewer outcomes.

4.      A measure of the typical burden of each outcomes (e.g. illustrative risk, or illustrative mean, on comparator intervention).

5.      The absolute and relative magnitude of effect measured for each (if both are appropriate).

6.      The numbers of participants and studies contributing to the analysis of each outcomes.

7.      A GRADE assessment of the overall certainty of the body of evidence for each outcome (which may vary by outcome).

8.      Space for comments.

9.      Explanations (formerly known as footnotes) (1).

Younas et al in 2021 wrote: “literature summary tables are not only meant to provide an overview of basic information (authors, country, purpose and findings) about included articles, but they should also provide detailed information about the theoretical and conceptual frameworks and the methods used in the included article (2)”.

Based on the reviewer’ comments, we summarized the tables:  

At the Table 1, we omitted “Obstetric history” column.

And also, we summarized the other tables too.

3.       The English text must be improved. I am fully aware of the difficulty encountered by scientists in non-English-speaking countries; nonetheless, the text of an article published in English must be properly written.

To help the Authors, I have made a series of suggestions, but it is the entire manuscript that requires review.

Thank you for this comment.

The manuscript was edited by a native English speaker.

4.       The manuscript mentions surgery for endometriosis, as well as surgery for appendicitis. I recommend that whenever surgery is mentioned, the Authors specify whether they refer to “laparoscopic” or “laparotomic” surgery, or to both.

Thank you for this valuable comment. Based on this comment we added this sentence:

All patients underwent surgery for removal of the appendix. Laparotomy or open appendectomy was used in the majority of cases.

Page 5, lines 14-15

And also type of surgery was written in Table 4.

5.       In the Abstract it is stated that a total of 1418 entries were subjected to “full-text screening”. If this were true, even if an article would take only 10 minutes to read, Authors would have spent more than 235 hours in screening their entries! In addition, under 2.2., Study selection, it is stated that “three trained authors screened titles and abstracts…”. This makes sense and the text of the Abstract should be modified, to coincide with what was done.

Yes. You are right, thank you for this valuable comment.

Abstract was revised.

6.       In the Introduction (page 2), mention should be made of whether any specific symptoms have been described in women with appendiceal endometriosis.

Thank you for this comment, it was added.

7.       Authors should be very careful in their description of the various groups to avoid misunderstanding, since the frequency of positive findings depends on how grouping is made. See Item 6 of the Specific Points.

Thank you for this appropriate comment.

According to this comments, we changed the groups, and merged groups.

Page 5, lines 2-7.

SPECIFIC POINTS

1.       Abstract: line 1: “This review was conducted to evaluate to presence…”. Lines 15-16: “women who were admitted…”; “The presence of appendiceal endometriosis was observed in 2.67%...”. Line 17: ”women who were admitted to the hospital…”. Line 21: “Conclusion: The presence of appendiceal endometriosis…”

Thank you, the manuscript totally was edited by a native English speaker.

     Introduction (page 2): Line three: What exactly do the authors mean when they mention endometriosis of “the genitalia”? Lines 5-6: “can be less frequently…”. Line 11: “must be distinguished…”. Line 18: “the preferred…”. Line 21: “are appendix-related complications…” Line 25: “several researchers…”. Lines 26-27: “disorders affecting women”; this is a vague statement: a brain tumor is a disorder that can affect a woman but does not require appendicectomy. Line 28: Benefits of carrying out an appendicectomy during surgery for…”. Lines 37-38: “in the present review we collected evidence for the topic”.

Introduction section was revised.

3.       Materials and methods. Page 2, first paragraph: I believe that authors meant that they have examined the “concomitant presence” of peritoneal and  appendiceal endometriosis. Page 3, line 3: “occurrence” of what? Line 14. It seems to me that what the Authors mean is “ acute appendicitis” (I am unaware of the existence of an “acute endometriosis”). Lines 20-21: “Conducted anywhere in world and published in English. No restrictions were applied to the search.

Method section was revised.

We appreciate your valuable comment.

4.       Results, page 3: Line 9 from the bottom: We initially identified 1418 publications…”. Line 8 from the bottom: What exactly is meant by “duplicate articles”? I presume that they were quoted more than once. Last two lines: were these 2 studies included?

It was revised.

“A total of 1418 publications, 555 of which were duplicate articles, were found in the various databases”

Page 3, lines: 40-41

5.       Study characteristics (page 4), line 1: “The present review includes 75 articles…”.

It was revised.

Page 5, line 1-2

6.       Item 3.2.1., page 6. Line 2: “who were admitted…”. This paragraph states that “appendiceal endometriosis was the most reported histological finding in these women”. There is something missing in this description because the text leads to the conclusion that in most cases of acute appendicitis, this is caused by endometriosis. This cannot be, since authors collected 44 cases over some 45 years, whereas every year there are thousands of cases of acute appendicitis that undergo surgery, as documented in Item 3.2.2. on page 9.

We revised the paragraph and title of this section:

Page 5, lines 10-16

7.       Item 3.2.3., page 10, Line 2: “reported on…”. Line 3: “The incidence of appendiceal endometriosis…”. Line 14: Instead of “while” it should be “On the other hand, the investigation by Pittaway et al concluded that...”. Line 16: I presume that authors mean that “the number of endometriotic foci…”.

It was revised.

8.       Item 3.2.4, page 11, Line 4: “According to these studies…”

It was revised.

9.       Discussion (page 13), first three lines: I am not sure of the meaning of this sentence: what is meant by the words “the high incidence of diseases in women with endometriosis”. Which diseases? Line 4: Obviously, a review is conducted to review a subject! The text needs improvement. Line 19: It is stated that the pathogenesis of appendiceal endometriosis involves retrograde menstruation through the Fallopian tubes. Since the appendix does not communicate with the peritoneal cavity, such a hypothesis needs further explanation. Lines 21-23: the sentence is incomplete. Lines 24.25: this sentence is also incomplete. Lines 28-31: I am not sure I understand why a histological diagnosis would depend on the “perception of the surgeon”. Line 38: The incredible range of frequency mentioned needs an explanation, even tentative.

Thank you for this comment.

Whole part of discussion was revised.

  1. Schünemann, Holger J., Julian PT Higgins, Gunn E. Vist, Paul Glasziou, Elie A. Akl, Nicole Skoetz, Gordon H. Guyatt, and Cochrane GRADEing Methods Group (formerly Applicability and Recommendations Methods Group) and the Cochrane Statistical Methods Group. "Completing ‘Summary of findings’ tables and grading the certainty of the evidence." Cochrane Handbook for systematic reviews of interventions(2019): 375-402.
  2. Younas A, Ali P. Five tips for developing useful literature summary tables for writing review articles. Evid Based Nurs. 2021 Apr;24(2):32-34. doi: 10.1136/ebnurs-2021-103417. Epub 2021 Mar 4. PMID: 33674415.

Round 2

Reviewer 1 Report

1. Thank you for an excellent and comprehensive revision of your manuscript.

2. The page numbering does not begin until page 13 on my copy. I cannot tell if that is when you started the numbering or if that is a Journal error. Please be sure your copy starts on page 1.

3. Adding "a mere 0.005%" conflicts with much of the rest of the paper. This is now reviewed for conflicts.

4. Using the various concepts of occurrence, prevalence, and incidents creates apparent conflicts. Please use 1) only one or 2) one, such as prevalence, for reference statistics when discussing the other two.

5. This paper includes that the incidence may be "a mere 0.005%," the prevalence is as high as 44.3%, and there is a "high prevalence of endometriosis." That is not statistically or verbally consistent. The mentions of a quote high prevalence", ranging from 1% to 44.3%, and "7.23 percentage "range, 1-44.3%)) are compatible with what I have seen. Please change the "it's incidences a mere 0.005%" to a phrase compatible with the range of 1% to 44.3% seen elsewhere in the article.

6. Please be consistent in your use of occurrence, prevalence, and incidence.

7. Samson (reference 101) discusses "endometrial type" tissue, not endometrium. The term "ectopic endometrium" is seen twice in the discussion. Change that to "endometriosis," "ectopic endometrial-like tissue," or Sampson's "ectopic endometrial type tissue."

8. There is a comment in response regarding "An international terminology for endometriosis, 2021." If you wish to use that, reference it. If they have "endometrial glands…," they need to update their definition. We now anticipate that endometriosis may be an endometrial, Mullerian rest, peritoneal stem cell, bone-marrow stem cell, other stem cell, or other derivatives. The endometrium is not the only candidate.

9. "Endometrium-like" is used once. Decide on the term you will use, such as "endometrial-like," "endometriotic," or "endometrial-type." Use the term you choose consistently.

10. Reference 30, Hori et al., for 0.005%, is an indirect reference to Collins, PMID 14098730, a 1963 summary of 1923 to 1963 (?) pathology. That may be useful in contrasting findings of the early and mid-1960s to those after the increased recognition that began in the 1980s. It is otherwise irrelevant. If you wish to keep it, add a discussion of the explosion in recognition that began with Goldstein's 1980, Jansen's 1986, and Moore's 1988 papers on recognition.

Goldstein DP, De Cholnoky C, Emans SJ. Adolescent endometriosis. J Adolesc Health Care. 1980 Sep;1(1):37-41. doi: 10.1016/s0197-0070(80)80007-6. PMID: 6458589.

Jansen RP, Russell P. Nonpigmented endometriosis: clinical, laparoscopic, and pathologic definition. Am J Obstet Gynecol. 1986 Dec;155(6):1154-9. doi: 10.1016/0002-9378(86)90136-5. PMID: 2947467.

Moore JG, Binstock MA, Growdon WA. The clinical implications of retroperitoneal endometriosis. Am J Obstet Gynecol. 1988, 158(6 Pt 1):1291-1298. doi: 10.1016/0002-9378(88)90359-6. PMID: 3381857.

11. "A comprehensive review of the literature" should include some or all of the discussions in Reference 30, Hori et al., of pain, vomiting, melena, the unclear relation to periodic menstruation, digestive symptoms, intestinal invagination, perforative peritonitis, the possibility of an incidental relationship with digestive symptoms, a relationship with periodical menstruation, symptoms even after menopause, the sensitivity of endometriosis, supportive care, medical therapy, and surgical treatment, and associations with cancer. Consider adding some or all of those.

12. Discuss the non-operative approach to appendicitis. Emile SH, Sakr A, Shalaby M, Elfeki H. Efficacy and Safety of Non-operative management of uncomplicated acute appendicitis compared to appendectomy: An umbrella review of systematic reviews and meta-analyses. World J Surg. 2022 May;46(5):1022-1038. doi: 10.1007/s00268-022-06446-8. Epub 2022 Jan 13. PMID: 35024922; PMCID: PMC8756749.

13. Endometrium-like is used once. Replace it with your choice of a standard term.

14. Endometrial is used 11 times. Two of those are correct as a citation to reference #101. Sampson JA. Intestinal adenomas of endometrial type: their importance and their relation to ovarian hematomas of endometrial type (perforating hemorrhagic cysts of the ovary). Archives of Surgery. 1922;5(2):217-80. are correct. The other none are suspect and may be misinterpreted. Clarify if those should be endometrial, endometrial-like, endometriotic, or the standard term you have chosen.

15. In "At this time, Pittaway et al. 3 suggested incidental appendectomy at the time of laparoscopy due to the high prevalence of pathologies in women 4 with endometriosis (87)," clarify the "pathologies."

16. Hori et al. (2021 discuss the controversy of "alimentary tract may be associated with digestive symptoms (e.g., pain, vomiting, and melena) remains controversial." "and whether digestive symptoms in these patients are related to periodic menstruation is unclear," Those are relevant, and a discussion should be added.

Author Response

Reviewer#1

Reviewer#1

1. Thank you for an excellent and comprehensive revision of your manuscript.

Thank you for your feedback.

2. The page numbering does not begin until page 13 on my copy. I cannot tell if that is when you started the numbering or if that is a Journal error. Please be sure your copy starts on page 1.

It was checked, page numbering in author copy is correct, maybe it is a system error. We apologize for this problem.

3. Adding "a mere 0.005%" conflicts with much of the rest of the paper. This is now reviewed for conflicts.

This figure was deleted.  

4. Using the various concepts of occurrence, prevalence, and incidents creates apparent conflicts. Please use 1) only one or 2) one, such as prevalence, for reference statistics when discussing the other two.

Thank you for this valuable comment. We revised it. And we chose prevalence, as all studies were checked and noticed prevalence is more correct term than other terms.

We highlighted all “prevalence” words in the manuscript.

5. This paper includes that the incidence may be "a mere 0.005%," the prevalence is as high as 44.3%, and there is a "high prevalence of endometriosis." That is not statistically or verbally consistent. The mentions of a quote high prevalence", ranging from 1% to 44.3%, and "7.23 percentage "range, 1-44.3%)) are compatible with what I have seen. Please change the "it's incidences a mere 0.005%" to a phrase compatible with the range of 1% to 44.3% seen elsewhere in the article.

Since we revised the introduction and discussion sections, therefore this percentage was deleted.

In result section we wrote the prevalence of this problem ranges between 1% to 44.3%.

Page 10, line 9.

6. Please be consistent in your use of occurrence, prevalence, and incidence.

Thank you for bringing it to our mind.

We used the word of “prevalence”.

Please refer the highlighted words in the whole manuscript.

7. Samson (reference 101) discusses "endometrial type" tissue, not endometrium. The term "ectopic endometrium" is seen twice in the discussion. Change that to "endometriosis," "ectopic endometrial-like tissue," or Sampson's "ectopic endometrial type tissue."

Thank you for this comment and subsequent comments regarding term of “endometrial” “endometrium –like”,

Based on the reference “An International Terminology for Endometriosis, 2021” (1) we chose “endometrium-like epithelium” and all those suspected term were changed and revised. We really appreciate the honorable reviewer’s attention and patience. 

We discussed Samson theory in discussion section.

Page 14, lines 12-3

8. There is a comment in response regarding "An international terminology for endometriosis, 2021." If you wish to use that, reference it. If they have "endometrial glands…," they need to update their definition. We now anticipate that endometriosis may be an endometrial, Mullerian rest, peritoneal stem cell, bone-marrow stem cell, other stem cell, or other derivatives. The endometrium is not the only candidate.

Thank you for this comment.

We defined endometriosis based on the term of the “An international terminology for endometriosis, 2021” reference and also cited that.

“Endometriosis is an estrogen-dependent disease defined by the presence of endometrium-like epithelium and/or stroma outside the endometrium and myometrium, usually with an associated inflammatory process (1).”

Page 2, lines 5-7

9. "Endometrium-like" is used once. Decide on the term you will use, such as "endometrial-like," "endometriotic," or "endometrial-type." Use the term you choose consistently.

Thank you for this comment. It was revised. We chose “endometrium-like epithelium” and converted all words to “Endometrium-like” based on the “An International Terminology for Endometriosis, 2021” and cited this refence in the first sentences of introduction.

Please refer to green highlighted words.

10. Reference 30, Hori et al., for 0.005%, is an indirect reference to Collins, PMID 14098730, a 1963 summary of 1923 to 1963 (?) pathology. That may be useful in contrasting findings of the early and mid-1960s to those after the increased recognition that began in the 1980s. It is otherwise irrelevant. If you wish to keep it, add a discussion of the explosion in recognition that began with Goldstein's 1980, Jansen's 1986, and Moore's 1988 papers on recognition.

Goldstein DP, De Cholnoky C, Emans SJ. Adolescent endometriosis. J Adolesc Health Care. 1980 Sep;1(1):37-41. doi: 10.1016/s0197-0070(80)80007-6. PMID: 6458589.

Jansen RP, Russell P. Nonpigmented endometriosis: clinical, laparoscopic, and pathologic definition. Am J Obstet Gynecol. 1986 Dec;155(6):1154-9. doi: 10.1016/0002-9378(86)90136-5. PMID: 2947467.

Moore JG, Binstock MA, Growdon WA. The clinical implications of retroperitoneal endometriosis. Am J Obstet Gynecol. 1988, 158(6 Pt 1):1291-1298. doi: 10.1016/0002-9378(88)90359-6. PMID: 3381857.

We appreciate your attention and patience.

Since we talked about the prevalence of “endometriosis in appendix” at the introduction section, therefore we deleted the mentioned reference (Hori et al., for 0.005%,) from this section to remove any doubts.

“Based on the reviewer comment which “It is otherwise irrelevant. If you wish to keep it, add a discussion of the explosion in recognition that began with Goldstein's 1980, Jansen's 1986, and Moore's 1988 papers on recognition.” We realized it is irrelevant. So, we deleted it from the manuscript. Since we deleted it, so there is no possible to discuss about it.

11. "A comprehensive review of the literature" should include some or all of the discussions in Reference 30, Hori et al., of pain, vomiting, melena, the unclear relation to periodic menstruation, digestive symptoms, intestinal invagination, perforative peritonitis, the possibility of an incidental relationship with digestive symptoms, a relationship with periodical menstruation, symptoms even after menopause, the sensitivity of endometriosis, supportive care, medical therapy, and surgical treatment, and associations with cancer. Consider adding some or all of those.

Thank you, based on this comment and following comments we revised the discussion section and considered 10 references of study by Hori et al.

Page 14. Lines 22-38

12. Discuss the non-operative approach to appendicitis. Emile SH, Sakr A, Shalaby M, Elfeki H. Efficacy and Safety of Non-operative management of uncomplicated acute appendicitis compared to appendectomy: An umbrella review of systematic reviews and meta-analyses. World J Surg. 2022 May;46(5):1022-1038. doi: 10.1007/s00268-022-06446-8. Epub 2022 Jan 13. PMID: 35024922; PMCID: PMC8756749.

Thank you for this comment. It was discussed.

Page 14, lines 32-38

13. Endometrium-like is used once. Replace it with your choice of a standard term.

Thank you.

Based on the “An international terminology for endometriosis, 2021” we used “Endometrium-like epithelium”

Please refer to green highlighted words.

14. Endometrial is used 11 times. Two of those are correct as a citation to reference #101. Sampson JA. Intestinal adenomas of endometrial type: their importance and their relation to ovarian hematomas of endometrial type (perforating hemorrhagic cysts of the ovary). Archives of Surgery. 1922;5(2):217-80. are correct. The other none are suspect and may be misinterpreted. Clarify if those should be endometrial, endometrial-like, endometriotic, or the standard term you have chosen.

Thank you for this valuable comment.

All “endometrial” words were checked and they were replaced by standard terms.

Based on the “An international terminology for endometriosis, 2021” we used “Endometrium-like epithelium”

Please refer to green highlighted words.

15. In "At this time, Pittaway et al. 3 suggested incidental appendectomy at the time of laparoscopy due to the high prevalence of pathologies in women 4 with endometriosis (87)," clarify the "pathologies."

It was revised.

Pathologies was “appendiceal endometriosis”

Page 12, line 1

16. Hori et al. (2021 discuss the controversy of "alimentary tract may be associated with digestive symptoms (e.g., pain, vomiting, and melena) remains controversial." "and whether digestive symptoms in these patients are related to periodic menstruation is unclear," Those are relevant, and a discussion should be added.

It was discussed.

Page 14, lines 22-30.

1. Thank you for an excellent and comprehensive revision of your manuscript.

Thank you for your feedback.

2. The page numbering does not begin until page 13 on my copy. I cannot tell if that is when you started the numbering or if that is a Journal error. Please be sure your copy starts on page 1.

It was checked, page numbering in author copy is correct, maybe it is a system error. We apologize for this problem.

3. Adding "a mere 0.005%" conflicts with much of the rest of the paper. This is now reviewed for conflicts.

It was revised and changed to 1%.

4. Using the various concepts of occurrence, prevalence, and incidents creates apparent conflicts. Please use 1) only one or 2) one, such as prevalence, for reference statistics when discussing the other two.

Thank you for this valuable comment. We revised it. And we chose prevalence, as all studies were checked and noticed prevalence is more correct term than other terms.

5. This paper includes that the incidence may be "a mere 0.005%," the prevalence is as high as 44.3%, and there is a "high prevalence of endometriosis." That is not statistically or verbally consistent. The mentions of a quote high prevalence", ranging from 1% to 44.3%, and "7.23 percentage "range, 1-44.3%)) are compatible with what I have seen. Please change the "it's incidences a mere 0.005%" to a phrase compatible with the range of 1% to 44.3% seen elsewhere in the article.

It was revised.

6. Please be consistent in your use of occurrence, prevalence, and incidence.

Thank you for bringing it to our mind.

We used the word of “prevalence”.

7. Samson (reference 101) discusses "endometrial type" tissue, not endometrium. The term "ectopic endometrium" is seen twice in the discussion. Change that to "endometriosis," "ectopic endometrial-like tissue," or Sampson's "ectopic endometrial type tissue."

Thank you for this comment and subsequent comments regarding term of “endometrial” “endometrium –like”,

Based on the reference “An International Terminology for Endometriosis, 2021” (1) we chose “endometrium-like epithelium” and all those suspected term were changed and revised. We really appreciate the honorable reviewer’s attention and patience. 

8. There is a comment in response regarding "An international terminology for endometriosis, 2021." If you wish to use that, reference it. If they have "endometrial glands…," they need to update their definition. We now anticipate that endometriosis may be an endometrial, Mullerian rest, peritoneal stem cell, bone-marrow stem cell, other stem cell, or other derivatives. The endometrium is not the only candidate.

Thank you for this comment.

We defined endometriosis based on the term of the “An international terminology for endometriosis, 2021” reference and also cited that.

Reference 1.

9. "Endometrium-like" is used once. Decide on the term you will use, such as "endometrial-like," "endometriotic," or "endometrial-type." Use the term you choose consistently.

Thank you for this comment. It was revised. We chose “endometrium-like epithelium” and converted all words to “Endometrium-like” based on the “An International Terminology for Endometriosis, 2021” and cited this refence in the first sentences of introduction.

10. Reference 30, Hori et al., for 0.005%, is an indirect reference to Collins, PMID 14098730, a 1963 summary of 1923 to 1963 (?) pathology. That may be useful in contrasting findings of the early and mid-1960s to those after the increased recognition that began in the 1980s. It is otherwise irrelevant. If you wish to keep it, add a discussion of the explosion in recognition that began with Goldstein's 1980, Jansen's 1986, and Moore's 1988 papers on recognition.

Goldstein DP, De Cholnoky C, Emans SJ. Adolescent endometriosis. J Adolesc Health Care. 1980 Sep;1(1):37-41. doi: 10.1016/s0197-0070(80)80007-6. PMID: 6458589.

Jansen RP, Russell P. Nonpigmented endometriosis: clinical, laparoscopic, and pathologic definition. Am J Obstet Gynecol. 1986 Dec;155(6):1154-9. doi: 10.1016/0002-9378(86)90136-5. PMID: 2947467.

Moore JG, Binstock MA, Growdon WA. The clinical implications of retroperitoneal endometriosis. Am J Obstet Gynecol. 1988, 158(6 Pt 1):1291-1298. doi: 10.1016/0002-9378(88)90359-6. PMID: 3381857.

We appreciate your attention and patience.

Since we talked about the prevalence of “endometriosis in appendix” at the introduction section, therefore we deleted the mentioned reference (Hori et al., for 0.005%,) from this section to remove any doubts.

Thank you for suggestion of these references, we liked that it was possible to compare different period times, but since the referees mentioned that the manuscript was long and suggested that it needed to be shortened, so we did not compare the periods anymore.

11. "A comprehensive review of the literature" should include some or all of the discussions in Reference 30, Hori et al., of pain, vomiting, melena, the unclear relation to periodic menstruation, digestive symptoms, intestinal invagination, perforative peritonitis, the possibility of an incidental relationship with digestive symptoms, a relationship with periodical menstruation, symptoms even after menopause, the sensitivity of endometriosis, supportive care, medical therapy, and surgical treatment, and associations with cancer. Consider adding some or all of those.

Thank you, based on this comment and following comments we revised the discussion section and considered 10 references of study by Hori et al.

Page 14. Lines 22-38

12. Discuss the non-operative approach to appendicitis. Emile SH, Sakr A, Shalaby M, Elfeki H. Efficacy and Safety of Non-operative management of uncomplicated acute appendicitis compared to appendectomy: An umbrella review of systematic reviews and meta-analyses. World J Surg. 2022 May;46(5):1022-1038. doi: 10.1007/s00268-022-06446-8. Epub 2022 Jan 13. PMID: 35024922; PMCID: PMC8756749.

Thank you for this comment. It was discussed.

Page 14, lines 32-38

13. Endometrium-like is used once. Replace it with your choice of a standard term.

Thank you.

Based on the “An international terminology for endometriosis, 2021” we used “Endometrium-like”

14. Endometrial is used 11 times. Two of those are correct as a citation to reference #101. Sampson JA. Intestinal adenomas of endometrial type: their importance and their relation to ovarian hematomas of endometrial type (perforating hemorrhagic cysts of the ovary). Archives of Surgery. 1922;5(2):217-80. are correct. The other none are suspect and may be misinterpreted. Clarify if those should be endometrial, endometrial-like, endometriotic, or the standard term you have chosen.

Thank you for this valuable comment.

All “endometrial” words were checked and they were replaced by standard terms.

15. In "At this time, Pittaway et al. 3 suggested incidental appendectomy at the time of laparoscopy due to the high prevalence of pathologies in women 4 with endometriosis (87)," clarify the "pathologies."

It was revised.

16. Hori et al. (2021 discuss the controversy of "alimentary tract may be associated with digestive symptoms (e.g., pain, vomiting, and melena) remains controversial." "and whether digestive symptoms in these patients are related to periodic menstruation is unclear," Those are relevant, and a discussion should be added.

It was discussed.

Page 14, lines 22-30.

Reviewer 2 Report

The revised text is definitely improved. I only have a few further comments as follow:

Page 1, line 38: The way this is written it gives the impression that there were complications “after 12 weeks”. A better sentence would be: during the 12 weeks follow-up no complications were reported.

Page 2, line 3: the word “inflammatory” should be omitted, since the situation is now explained in lines 5-6.

Page 3, line 25: the word “exclusively” should be omitted.

Page 5, Line 15: I do not understand the meaning of sentence “Laparotomy or open appendectomy was used in the majority of cases”. What is the difference between “laparotomy” and “open”? If these 2 modalities were used “in the majority of cases”, which other techniques were used?

Page 10, Item 3.2.2.: The expression “endometriosis of the appendix” is used 8 times. Perhaps it may be better to use an abbreviation.

Page 13, lines 16-17: It may be better to say: “Periodic menstrual bleeding in the ectopic tissue may trigger acute appendicitis”. Line 17-18: The “ectopic transplantation theory” presupposes that endometrial debris may implant on the peritoneal surface of the appendix and then penetrate inside. Is this what the hypothesis implies? Lines 27-29: The main question here is whether subjects with endometriosis of the appendix have “peritoneal endometriosis”, not just ovarian endometrioma. What information is available on this subject? Line 44: “surgery”!

Author Response

Reviewer#2

Reviewer#2

The revised text is definitely improved. I only have a few further comments as follow:

Page 1, line 38: The way this is written it gives the impression that there were complications “after 12 weeks”. A better sentence would be: during the 12 weeks follow-up no complications were reported.

Thank you. It was done.

Page 1, lines 39-40

Page 2, line 3: the word “inflammatory” should be omitted, since the situation is now explained in lines 5-6.

It was deleted.

Page 2, line 2.

Page 3, line 25: the word “exclusively” should be omitted.

It was deleted.

Page 3, line 26.

Page 5, Line 15: I do not understand the meaning of sentence “Laparotomy or open appendectomy was used in the majority of cases”. What is the difference between “laparotomy” and “open”? If these 2 modalities were used “in the majority of cases”, which other techniques were used?

It was a type error, it was revised. And also, we added “and laparoscopy in the rest” to show the other techniques were used.

Page 5, line 14.

Page 10, Item 3.2.2.: The expression “endometriosis of the appendix” is used 8 times. Perhaps it may be better to use an abbreviation.

Since the referees in the first round of review suggested us to use endofound terminology”, so we searched to find an approved abbreviation for “endometriosis of the appendix” to use, but there was not suggestion. So, with all respect to honorable refugee we kept those words.

Page 13, lines 16-17: It may be better to say: “Periodic menstrual bleeding in the ectopic tissue may trigger acute appendicitis”. Line 17-18: The “ectopic transplantation theory” presupposes that endometrial debris may implant on the peritoneal surface of the appendix and then penetrate inside. Is this what the hypothesis implies? Lines 27-29: The main question here is whether subjects with endometriosis of the appendix have “peritoneal endometriosis”, not just ovarian endometrioma. What information is available on this subject? Line 44: “surgery”!

Based on these comments, discussion section was revised.

Page 14, lines 11-38

The revised text is definitely improved. I only have a few further comments as follow:

Page 1, line 38: The way this is written it gives the impression that there were complications “after 12 weeks”. A better sentence would be: during the 12 weeks follow-up no complications were reported.

Thank you. It was done.

Page 2, line 3: the word “inflammatory” should be omitted, since the situation is now explained in lines 5-6.

It was deleted.

Page 3, line 25: the word “exclusively” should be omitted.

It was deleted.

Page 5, Line 15: I do not understand the meaning of sentence “Laparotomy or open appendectomy was used in the majority of cases”. What is the difference between “laparotomy” and “open”? If these 2 modalities were used “in the majority of cases”, which other techniques were used?

It was a type error, it was revised. And also, we added “and laparoscopy in the rest” to show the other techniques were used.

Page 10, Item 3.2.2.: The expression “endometriosis of the appendix” is used 8 times. Perhaps it may be better to use an abbreviation.

Since the referees in the first round of review suggested us to use endofound terminology”, so we searched to find an approved abbreviation for “endometriosis of the appendix” to use, but there was not suggestion. So, with all respect to honorable refugee we kept those words.

Page 13, lines 16-17: It may be better to say: “Periodic menstrual bleeding in the ectopic tissue may trigger acute appendicitis”. Line 17-18: The “ectopic transplantation theory” presupposes that endometrial debris may implant on the peritoneal surface of the appendix and then penetrate inside. Is this what the hypothesis implies? Lines 27-29: The main question here is whether subjects with endometriosis of the appendix have “peritoneal endometriosis”, not just ovarian endometrioma. What information is available on this subject? Line 44: “surgery”!

Based on these comments, discussion section was revised.

Page 14, lines 11-38
